# *'You don't have to sleep with a man to get how to survive'*: Girl's perceptions of an intervention study aimed at improving sexual and reproductive health and schooling outcomes

**Linda Mason**[1]*, **Garazi Zulaika**[1], **Anna Maria van Eijk**[1], **Eunice Fwaya**[2], **David Obor**[2], **Penelope Phillips-Howard**[1], **Elizabeth Nyothach**[3]

**1** Liverpool School of Tropical Medicine (LSTM), Liverpool, United Kingdom, **2** Ministry of Health, Siaya County, Kenya, **3** Kenya Medical Research Institute (KEMRI), Centre for Global Health Research, Kisumu, Kenya

* Linda.Mason@lstmed.ac.uk

**Data Availability Statement:** This study was conducted with approval from the Kenya Medical

## Abstract

In sub-Saharan Africa, girls suffer from high rates of morbidity and mortality, enduring high exposure to sexual and reproductive health harms. Staying in school helps protect girls from such harms. Focus group discussions were conducted in a rural, impoverished area of Kenya with adolescent girls participating in a 4-arm cluster randomised controlled trial, evaluating menstrual cups, cash transfer, or combined cups plus cash transfer against controls. To explore girls' perceptions of how trial interventions affected their SRH risks and schooling, semi-structured discussions were held at baseline, midline, and study end. Data was explored using thematic analysis. At baseline there were no discernible differences between the 4 intervention groups regarding their perceptions of relationships with boys/men, and difficulties attending or remaining in school. Midline and endline discussions found that narratives from those receiving cash transfer only, or alongside a cup were similar; girls noted fewer pregnancies and less school dropout, attributed to the cash transfer reducing the need for transactional sex. Lower absenteeism was reported by the cup only group, with perceived minimal effect on pregnancy and dropout. Girls in control and cup only groups described feeling valued through inclusion, benefitting from puberty and hygiene education. Although seemingly having little effect on reducing pregnancy or dropout, these inputs reportedly empowered girls, whilst cash transfer girls were emboldened to refuse male sexual advances. Girls noticed benefits from trial interventions, with a reduction in transactional sex and resulting pregnancy impacting on school dropout, or reduced menstrual related absenteeism. Education and study inclusion were perceived as important. Future programmes should consider alleviating material deprivation which prevents girls from attending or performing at school through schemes such as cash transfer, alongside hygiene and education packages. This will empower girls to refuse unwanted sex and understand risks, in addition to motivating academic achievement and school completion.

**Trial registration:** ClinicalTrials.gov NCT03051789.

Research Institute (KEMRI) Scientific and Ethics Review Unit (SERU), which requires that data be released from any KEMRI-based Kenyan studies (including de-identified data) only after their written approval for additional analyses. In accordance, data for this study will be available upon request, after obtaining written approval for the proposed analysis from the KEMRI SERU. Their application forms and guidelines can be accessed at https:// www.kemri.org/seru-overview. To request these data, please contact the KEMRI SERU at seru@kemri.org.

**Funding:** This study is funded by the Joint Global Health Trials Initiative (UK-Medical Research Council/Department for International Development/ Wellcome Trust/Department of Health and Social Care, grant MR/N006046/1). PPH received the award, grant MR/N006046/1. We would like to thank Mooncup UK for providing the cups at discounted price, and Equity Bank for their support with cash transfer. The funders have no role in the design of the study, the collection, analysis, and interpretation of data, or in writing the manuscript.

**Competing interests:** The authors have declared that no competing interests exist.

## Introduction

Adolescent girls and young women (AGYW) in low- and middle-income countries (LMICs) suffer a disparate proportion of morbidity and mortality, especially in sub-Saharan Africa (SSA) where they endure twice the mortality rate of AGYW in any other region [1, 2]. Of particular concern is the high exposure to sexual and reproductive health (SRH) harms. In SSA these include HIV, (where adolescent girls have twice the rate of new infections compared to adolescent boys), sexually transmitted infections (STIs) and reproductive tract infections (RTIs), female genital mutilation, unsafe abortion and adolescent pregnancy [3–5]. Babies born to adolescents are more likely to be preterm, underweight and have a higher risk of dying in infancy [6].

Protecting girls from SRH harms helps reduce gender inequity and prevents the cycle of poverty and ill-health being passed down to the next generation. Staying in school is one such way to protect girls, with several mechanisms interlinking to facilitate this. Girls who remain in education have a greater likelihood of using contraception, delaying marriage, having a later first birth and of not being exposed to intimate partner violence [7]. Schoolgirls are more likely to report fewer sexual partners, with less age disparity between them, and less frequent sex [8–10]. In addition to improved health and wellbeing, education links with self-agency. A report published by the World Bank [7] states '*Fundamentally, a lack of education disempowers women and girls in ways that deprive them of their basic rights'.* Girls and women who have been educated have more voice in decision-making at the household and employment level, with greater potential for employment and higher wage-earning capacity [7].

The introduction of free primary education across SSA has enabled most girls to complete primary school. However, poverty and gender inequity contribute to low rates of secondary school enrollment. According to the World Bank many girls who enroll each school year drop out before the year ends; indicators estimate just four in ten girls complete secondary school in SSA [2]. Recent legislative changes within many SSA countries now permit girls to remain in school if pregnant, yet social and cultural mores may discourage attendance among girls who are visibly pregnant, or from re-enrolling as a mother ([11]). A recent study in Kenya reported that pregnancy drove school drop-out which in turn led to early marriage ([12]). Although in many SSA countries, including Kenya, primary education is free the current study setting, levies such as enrollment fees, together with material requirements (e.g. textbooks, uniforms, lunches, transport) leave students unable to complete secondary education [13, 14]. A complex interplay of factors alongside financial pressure drives school dropout [15]. In Nigeria, Uche [16] attributed 89% of dropout to poverty and 74% to pregnancy. A recent study in Uganda estimated over half of secondary school dropouts were due to pregnancy, with a quarter due to early marriage [17]. A recent Kenyan study reported that of 53 pregnant schoolgirls aged 14– 16 years, all but two dropped out and just 5 returned to school after delivering [12]. Also, in western Kenya, Oruko et al.'s qualitative study [18] reported pregnancy as the most commonly stated reason for dropout. Menstruation and household labour were also frequently cited. The authors concluded that poverty led to a cascade of events often ending in pregnancy and / or dropout.

Girls in rural Kenya report engaging in transactional sex to acquire necessities, including to fund their schooling needs for example monies to pay for fees, purchase uniforms and acquire sanitary pads [19–21]. Paradoxically, it is likely that transactional sex further contributes to unintended pregnancy, high rates of HIV and other STIs, leading to school dropout. Thus contributing to the cycle of poverty rather than keeping girls in school [18, 19, 22].

School absence has been strongly linked with poor menstrual hygiene management (MHM) in qualitative studies. Reasons include lack of suitable absorbents, fear of leakage and

period teasing [23, 24]. However, less robust evidence has been generated from quantitative studies, [25] with only marginal improvements seen when piloting menstrual products [26–28]

This paper comprises qualitative analysis of a series of FGDs which were conducted as part of a wider randomised controlled trial (described below) to explore girls' perceptions of the effect that trial interventions had on their relationships with boys and men, and on school dropout and absence. The FDGs were designed to also allow for any outcomes of importance to the girls themselves to emerge. The trial interventions chosen for this study were cash transfer and / or menstrual cups.

## Methods

### Ethics statement

The protocol, participant information sheets, the informed parent consent and participant assent documents, and tools were reviewed and approved by the Scientific and Ethics Unit (SERU; #3215) and the Liverpool School of Tropical Medicine, Liverpool (LSTM; #15–005).

### The main trial study design

The main study was a 4-arm cluster randomised controlled trial conducted in 96 secondary schools in rural western Kenya between 2017 and 2021, (described in detail elsewhere [29]; Schools were selected if they were day schools, were female students, and had approval from the principal, and were excluded if they only accepted full-board students, were special needs, or did not include female students. Schoolgirls were eligible if they were in the eligible study schools, in the selected school years (grades), had reached menarche, were not visibly pregnant, had no disability precluding participation, had parental consent and gave personal assent. All eligible secondary schools were randomised into one of four intervention arms: (1) menstrual cup arm all girls were provided with one Mooncup®, a reusable bell-shaped receptacle inserted to collect menstrual flow, together with training in safe cup use and care; (2) cash transfer (CT) arm (KES1500 [~$15]) pocket money issued each school term through bank cards conditional upon 80% or greater attendance in the previous school term as measured using school registers. Financial literacy training was also provided; (3) cups and cash combined intervention arm, with cup use and financial literacy training; (4) control arm i.e. 'usual practice'. Girls in the control arm and the cups only arm were provided with handwash soap each term. All participants received puberty and hygiene education at the start of the study. The interventions were chosen on the basis of prior research which has demonstrated that provision of cash transfer can impact positively on school indicators and protect against some STIs [10, 30], whilst our previous feasibility study found the menstrual cup can reduce rates of reproductive tract infections and STIs [31]). Furthermore, girls provided with a cup who had experienced using sanitary pads, appeared to express a preference for them over pads [32]. The primary outcome of the trial was a composite of HIV, HSV-2, and school dropout incidence by trial end, with a basket of SRH and schooling measures as secondary outcomes.

### Study setting

The study was conducted in Siaya County, an impoverished area in rural western Kenya, with a population comprising largely of subsistence farmers and fisherfolk [33] who are of the Luo ethnic group. Within this region, females suffer high morbidity and mortality with higher rates than the national average of HIV, TB, maternal mortality [34] and gender-based violence

[35]. A recent study reported adolescent school girls are frequently harassed for sex and face high risk of pregnancy and HSV2 [36].

## Inclusion criteria

Participants needed to reside within the study area, attend secondary day school in the study area, have a history of established menses and have no disability that would prevent full participation. Parental or guardian consent along with girls' own assent was also a requirement.

## Current qualitative study

A qualitative design was adopted comprising 24 focus group discussions (FGDs) among participating girls. FGD were chosen to maximise informal conversation between participants which we believed would provide data-rich contextual information in comparison to structured survey instruments or individual interviews. The FGDs were conducted at three different time points across the study. At baseline (pre-intervention) we explored girls' perceptions of school, alongside risk factors and reasons for drop-out. We concentrated on issues found in the literature to influence schooling and drop-out, including relationships and menstruation. At midline (approximately one year after receiving the intervention) and endline (approximately 2 years after receiving intervention) we explored issues around girls' perceptions of any impact the interventions had on their schooling and drop-out, again delving into potential influence on relationships and menstruation. Randomisation of the intervention took place at school level, so all eligible girls in the selected grades in that school received the same intervention provided that they met the criteria for study inclusion. Girls in the FGD were invited because of the intervention their school was allocated to. All participants in a cup group (with or without CT) were given a menstrual cup, but may not have used it. Similarly, girls in the CT group may not have received all of their monies as receipt was conditional upon 80% attendance in the previous term. The same applied for the combined CT / cup group. We only invited participants to an FGD according to intervention group, not according to receipt or usage. We undertook two FGDs within each study arm, a total of eight FGDs per time point. Simple random sampling of schools in each intervention arm was used to select schools for each series (baseline, midline, endline). Schools were not selected more than once. Within the selected schools, all trial participants were invited to attend the FGD. We aimed for 8–12 participants per group (with a quorum of 6) to balance diversity of opinions without overwhelming participants' ability to contribute in a group situation. If more than 12 girls volunteered, simple random sampling was used to select participants.

The study design allowed for additional FGD to take place should saturation not be reached; however, sufficient data were collected. (See S1 Fig)

**Tools.** A semi-structured FGD guide was developed by the research team following discussion of topics necessary to achieve the study aim (see S1–S3 Tables). Within topic headings, questions and prompts were drafted to guide the two moderators (acting as moderator and notetaker in rotation). The moderators were consulted during final drafting of the guide to incorporate their local knowledge and views, and to ensure they understood the rationale and objectives of the FGD. The moderators were young Kenyan women, who understood the local vernacular and were fluent in Dholuo, Kiswahili and English, thus facilitating free dialogue and minimising positionality affects.

## Procedure

Participants agreed to a series of ground rules such as the need for confidentiality, respect for each other's opinions, and it was emphasised that there were no 'right' or 'wrong' answers. An

ice breaker was introduced at the start of each session to put participants at ease and encourage them to be vocal. We stated they did not need to raise their hand and wait for permission to speak (as is customary in Kenyan schools) as this was '*their*' discussion. Participants were given a sequential number and asked to state this prior to speaking. Discussions were audio-recorded, and conducted in English, Kiswahili or Dholuo (sometimes a mixture), according to group preference. Notes were made to capture main points, group dynamics and non-verbal gestures. Discussions were transcribed verbatim by the moderator or note-taker with Luo/Kiswahili translated into English.

## Analysis

Thematic analysis was chosen [37] because we had no pre-conceived theory to test and wanted a flexible system allowing us to compare and contrast findings both across study arms and over time. A combined deductive and inductive approach was applied to obtain information relevant to the research aim, while also allowing for information pertinent to the participants themselves to emerge. The transcripts, imported into Nvivo software package v9, were read and re-read alongside the notes made during the FGDs. This facilitated familiarity with the data, checking for accuracy, and provided an overview of emerging themes, thereby indicating the first level codes to apply. Relevant data were assigned to one or more codes, with second level coding classifying the data into a framework of themes and subthemes. The FGDs covered topics wider than the research aim for the present paper, so from the framework we specifically chose the codes that would help us meet this aim. These covered schooling and outcomes, (drop-out, pregnancy, menstruation) as well as relationships and allowed us to compare and contrast across study arms as well as over the 3 timelines. A draft narrative was prepared, linking themes and subthemes with illustrative quotes. The draft was revisited alongside original transcripts to check that context and magnitude of findings were not misunderstood or exaggerated, and to interrogate whether alternative explanations fit the data better. Throughout the analysis and drafting of the paper, the first author conferred with the research team to ascertain language and described behaviours were understood and interpreted appropriately. With familiarity of the transcripts and notes, the team also ensured that the narrative appeared to represent the transcript data.

## Results

Two hundred and thirty one girls participated in total: This comprised 79 at baseline, 77 midline and 75 at endline with no girl participating in more than one FGD (See Table 1 for Group composition). For quotes, an assigned code defined the school: B (baseline), M (midline) or E (endline) to signify when the FGD took place. P [number] was used to represent the individual participants with Px denoting missing participant number. The results are presented firstly by baseline findings to illustrate the views and experiences of girls prior to receipt of intervention, whilst findings from mid and endline are combined due to minimal differences in girls' perceptions across these 2 time periods.

### Pre- intervention (baseline)

Girls discussed the following themes: causes of school dropout, drivers of absenteeism and nature of relationships with males including transactional sex, multiple partners and consequences of pregnancy.

At baseline there were no discernible differences in intervention group opinion. Most girls spoke positively about the value of schooling and the opportunities that it gave them, compared to past generations.

**Table 1. Focus group composition.**

| School | When | Intervention | Time (minutes) | No. participants | Age range (years) |
|---|---|---|---|---|---|
| 1 | Baseline | Pre-intervention | 80 | 11 | 16–18 |
| 2 | Baseline | Pre-intervention | 55 | 7 | 15–17 |
| 3 | Baseline | Pre-intervention | 80 | 12 | 17–18 |
| 4 | Baseline | Pre-intervention | 56 | 10 | 16–17 |
| 5 | Baseline | Pre-intervention | 68 | 9 | 16–18 |
| 6 | Baseline | Pre-intervention | 75 | 8 | 15–17 |
| 7 | Baseline | Pre-intervention | 38 | 11 | 16–17 |
| 8 | Baseline | Pre-intervention | 60 | 11 | 15–18 |
| 9 | Mid | Control | 85 | 6 | 17–18 |
| 10 | Mid | Control | 80 | 11 | 16–23 |
| 11 | Mid | Cup | 47 | 9 | 17–19 |
| 12 | Mid | Cup | 68 | 10 | 16–18 |
| 13 | Mid | Cash | 68 | 12 | 16–18 |
| 14 | Mid | Cash | 75 | 11 | 16–18 |
| 15 | Mid | Cash Cup | 95 | 9 | 16–18 |
| 16 | Mid | Cash Cup | 60 | 9 | 16–19 |
| 17 | End | Control | 70 | 9 | 18–20 |
| 18 | End | Control | 60 | 9 | 18–20 |
| 19 | End | Cup | 60 | 11 | 17–20 |
| 20 | End | Cup | 65 | 7 | 18–20 |
| 21 | End | Cash | 70 | 12 | 17–19 |
| 22 | End | Cash | 70 | 8 | 18–20 |
| 23 | End | Cash Cup | 65 | 10 | 19–21 |
| 24 | End | Cash Cup | 58 | 9 | 18–20 |

*'We go to school so that we may have a better life in future. I mean, so that we do not be like our parents, who are struggling to pay our school fee, but still they are overwhelmed. So we feel we need to try so that we become better than them'. (1B P8)*

Perceived causes of school dropout were deemed to be multi-factorial, often a cascade of events leading up to eventual dropout. Girls believed pregnancy and poverty—in particular the lack of school fees—were the key drivers, with menstruation also frequently mentioned as a contributor, *with one girl highlighting that if visible leakage of menstrual blood occurred to her it would seem so shameful she doubted she would be able to return to school.* A couple of girls suggested that gender issues such as a lack of parental support for girls' education may contribute; others indicated that poor academic performance or punishment often led girls to leave school. Peer pressure was also cited as a cause of drop-out, particularly if peers were earning money.

*'My parent will say, you are very stupid like your own people [paternal family], is that what you want to do in school! Its better I stop wasting my money, you better go get married because you are wasting my money'! (3B P1)*

*'She will drop school because most parents when they see you pregnant they will refuse to pay your school fees, they will tell you go to the person responsible for your pregnancy. So it will force you to drop because your fees cannot be paid anymore. They say you never wanted education'. (8B Px)*

'*One person then eventually faces you and asks, "kindly check on yourself, check [the blood on] your back". Once someone tells you that, almost the whole class will become aware of your situation. So when you finally get out of that school, you will never come back. Even me I can never come back to school*'. *(3B P12)*

Causes of absenteeism also appeared to be multifactorial, but the two major reasons cited were menstruation and chores. Typically, the girls described being very tired after a day's schooling followed by doing chores, often until late, and then having to do homework. For some, the chores started again in the morning before they left for school.

'*For me it can make you miss school because when am back home in the evening, I'm given a lot of chores that I could not manage to do homework or even touch my books to study, completing my homework will be impossible and then if I remember coming to school and I didn't finish my homework, I think the teacher will punish me and so I decide to miss school*' (4B P6)

There was no perceptible difference between the views expressed by intervention groups regarding the nature of their relationships with boys or men. Girls' relationships with class-mates or boys of similar ages were often caring and took the role of friendship, emotional or academic support, but were not often described as sexual relationships. Some girls were quick to point out classmates were friends as opposed to '*lovers*'.

'*You see here in school, like me I have a boyfriend, not even boyfriend, boyfriends, but they are not my lovers. Like in class you sit with them. . . . . . . . ..but he is not my lover*'. *(1B P6)*

The following excerpt was atypical of the conversations around peer relationships but demonstrated acknowledgement from a minority of girls that they had boyfriends for sexual pleasure. Noticeably, these relationships seemed to be alluded to in the context of male peers rather than emerging when asked about relationships with older men.

'*And then we also have this issue of hormones, let's say my hormones are high. I mean I just feel I need to have sex; I will be forced to call my boyfriend and tell him I am coming*'. *(3B P9)*

The relationships girls described with older men were not seemingly for love or affection, or even sexual gratification. These were rarely mentioned but instead were transactional arrangements which endowed an element of status amongst their peers because of the material rewards they could reap. Often described as '*sponsors*', these men might provide gifts, pay school fees, or provide cash for clothes, sanitary pads, food or luxuries. These older men also might be married and have families of their own. Girls understood the inevitability of the reciprocal arrangement, i.e., that they paid for receiving money or gifts by having sex.

'*No girl will go to a poor man. . . . . . ..they love money and girls are after money*'. (1B Px)

'*I can say some girls normally have [a] "sponsor", he will provide everything to the girl, so when someone tells you they have sponsor she is seen to be in a high class, so this can lure other girls to try and get one and it may lead you astray*'. *(6B Px)*

Boda boda (Motorbike taxi drivers) were often cited as partners in sexual transactions, but seemingly not recognised as having a relationship role such as '*boyfriend*' '*lover*' or '*sponsor*'. Instead, their role was functional, providing lifts to and from school which the girls accordingly paid for with their body.

'*Yes, you find that the boda boda man will be giving you lifts and you do not pay, and this is true, you don't pay. And maybe he has given you lift and when you reach the school gate he gives you money for escort. In the evening he comes to pick you. By weekend you will pay!*' (3B P6)

Some girls chose to have multiple partners to secure a range of requirements, maintaining one or two for love or affection.

'*We normally say that you select five isn't it*'? P6

'*Yes*'. [Chorus]

'*You select five, you choose four, you admire three, love two then you trust one*'. P6 (all 1B)

Girls were aware and scathing of the inevitability that they would be dropped immediately should they become pregnant. This was irrespective of whether they were in a 'relationship' with a boyfriend or sponsor.

'*What the men always want is how to sleep with you. They will only feel you are beautiful before they sleep with you....*"*come I will give you this and that*". *He will only provide for the first few days. The next day after sleeping with you, maybe he will make you pregnant. He will tell you,* "*Hey! My dear, take your luggage to your home*"' *You will be forced to leave school*'. (3B P1)

A few girls insinuated that some parents were aware of their having transactional sex but mostly turned a blind eye to any new purchases and didn't question where or how they were obtained. Some described how relatives actively reminded them they were old enough to find ways of obtaining monies for themselves, or even financially supporting their household.

'*This depends with the parents because, I have witnessed such parents who when their daughter stays for a week without visiting her boyfriend, the mother will ask, what is wrong with you, how are we eating*'? (5B P13)

## After interventions

In contrast to the baseline discussions, mid and end of study narratives demonstrated some differences between groups that girls perceived were due to intervention effects.

## The control group

Generally, the girls were positive about being in the trial. Some said that the puberty and hygiene education increased their confidence as they knew how to keep clean and not smell, particularly during their menses, and were grateful for the receipt of soap. They were happy to know their HIV status, although some disliked having a blood test. Many were positive about the guidance and counselling received during the education package and wanted to cascade the teaching to their peers. For some, it made them feel worthy of investment.

'*In this study generally it's like we are enjoying a certain protection because we are being taught on how to protect ourselves, and we are also taught on several virtues*' (10M P10)

'*In the past the boys used to complain the girls smell. . .there were so many complaints in the school that girls do not take bath during their monthly period and they were smelling, but nowadays such cases are very rare*' 9M P4

Attitudes differed somewhat regarding whether study participation had shaped their relationships with boys and men. Whilst some felt it had little or no effect, others suggested changes arose from increased drive and confidence, stemming from the puberty and education element, which empowered some girls to refuse unwanted advances.

'*Now me in this [study] group we were taught if a boy wants you, you either say yes or no, and when you say "no" you mean it–and a strong one! Me, in case I have been seduced by a boy in our class I have to say "no" and stand straight to say "no". Me, this [study] has helped me so much, when I say "no", it's a no'.* (17E Px)

A minority of girls proffered that there might be some slight reduction in school dropout or absence. This was attributed to the puberty and education element, as illustrated by extracts below.

'*the education we are given, we are being encouraged to stay in school, that our studies are more important so we are always motivated we cannot drop out of schoo*l' (18E P8) with a colleague adding '*then it is also giving us ways of protecting ourselves from early school dropout or early marriages, or engaging in premature sex that leads to early pregnanci*es' (18E P7)

Ultimately, however, despite the support received to assist girls in negotiating relationships, their material needs remained unmet. Consequently, many participants described having very little recourse other than to continue trading sex for money or gifts. Similarly, attendance rates were seen as staying constant, with girls frequently reporting missing school during their menses due to a lack of pads. A lack of school fees and books also contributed to school absence, at least until funding could be found, either from parents, or through engaging in transactional sex if parent-guardians were unable or unwilling to meet these costs.

P6: For me, I haven't seen any difference because we have said that they engage in sex even to get money even to buy pads an since we started this study, there is nothing that we have been provided with which can use to purchase other items that we need'. (17E P6)

'*Yes, like me for sometimes when the school fee is not there, I was forced to drop out [temporarily] because I had no one to support me in paying fee, so I went back home, stayed with my grandmother. She is not earning enough for her to be able to meet my needs, so I told her to stop paying fee because when she decides to pay my fee we are struggling in the house' (10M Px)*

## Girls in the cup only group

Some girls appreciated knowing their HIV status, although one girl went further stating '*it has helped us to continue abstaining to maintain being negative'* (11M P4) Again, there was acknowledgement that being counselled and encouraged through the puberty and education package helped to provide skills and a degree of confidence to negotiate with boys and men. Importantly it also reportedly heightened awareness that girls' education was as valuable as that of boys according to a couple of participants.

*'Aaah to me this [study] has really taught us on confidence and being aware as a girl you are important in the society so when they come they usually encourage us to work hard, that education is not meant for only boys and also us like girls we can make it because many girls have made it to the universities so they always encourage us to study hard'.* (20E P3)

'*Ok, according to me, I can say that it has helped me to respect myself because since I joined (the study) I know I am a girl of substance, I know what I want in future and that's why I have said that this organisation has helped us'* (19E P6)

Many girls expressed overwhelming gratitude in receiving the cup. A few expressed a process of familiarization wherein with time, persistence, or repeated training they came to use it regularly; a minority of girls in all four cup groups however chose not to use it. Users valued not needing to find ways to obtain monies each month to purchase sanitary pads. The majority opinion was that having a cup led to improved school outcomes. Many spoke of improved confidence and concentration through removal of the uncertainty or shame associated with leaking. Attendance was believed to have improved, with no need to stay at home through a lack of absorbent. It was also felt to diminish financial burden, thus reducing the need for transactional sex, either in terms of number of partners, or number of sexual acts. However, the consensus was that the need for other necessities remained, and transactional sex still provided a means to that end. Thus, the cup had minimal impact on school dropout because it was only part of the solution to their financial needs.

'*Since we have been having mooncups even if you are attending [*having a period] *and the teacher asks a question or ask you to go do a sum on the board, you will just be confident, so even if it is answering a question you will stand comfortably and answer the questions asked, and the more you answer the questions the more you learn and the more your performance improves'* (20E P5)

'*The moment we have mooncup at least half the load is reduced'* (12M Px)

'*I think the number of girls engaging in boy/girl relationships has reduced because most of the relationships are for the help during the buying of the sanitary pads'* (11M Px)

'*To me, I never lack problems, pad part is solved, then sometimes you find some important personal things like it will force you to buy, bra, panties, they are worn out. I stay with a parent who is male, when you tell him he tells you he doesn't have'* (12M P1)

## Girls in the CT only group

Conversations among girls receiving CT focused far less on the effect of the puberty and education training, compared with girls in the control and cup groups, despite receiving the same teaching session. Some girls mentioned having increased confidence, particularly when communicating with boys, but did not necessarily attribute this to the education. Some remarked that boys, and men, including some *boda-boda* riders, were less likely to approach them being somehow aware these girls had money of their own and would not need to trade. It was not apparent whether, or to what degree, these girls felt that the increase in confidence arose directly from the motivation and education from being in the study, having monies to purchase necessities (particularly sanitary pads, which appeared greatly important to participants), or boys' and men's reaction to their having monies.

'*Since we started these lessons, nowadays like me I can just approach a boy and look at him right in the face. So if he tells me his stories I just stop him once and I tell him that I do not want him and I don't feel like being with him and he goes on his way*' (14M Px)

'*Since you started supporting us with the money boys have really been avoiding us because. . . . . .boys just know that if I approach her, she has more money than we so there is nothing I will tell her that she will listen to, she will just ignore my words, so he decides to keep off*' (22E Px)

'*At the stage there if you pass the boda boda people now. . .every time you pass by they don't call you as they used to do before. It is now better because they are afraid to do so. (21E P5) 'Mmmhh. . .What has made them afraid'?(M) ' They know that so and so can now depend on herself even if she passes here'(21E P5).*

There was some mention made, not apparent in the control or cup group discussions, that girls were now independent rather than having to rely on boys, sponsors, or their parents or guardians. Those girls joyously spoke of becoming '*proud.*'

'*For now if a boy in form 4 approached me I will confidently tell the boy there is no good thing that he can give me that I don't have. So I am proud of myself*' (21E P2)

Most girls voiced their perception that having cash reduced the need for transactional sex, with some girls stating that it stopped this behaviour altogether. However, it did not appear to negate the necessity, or inclination completely for all girls. A few acknowledged less reliance on boys / men but admitted they (or '*others*') still had financial needs or desires to be fulfilled by one or more boyfriend or sponsor.

'*I see that the money has really helped us because as girls if we would not be receiving the support from the project some of us would have gotten pregnant, ran out of school, so I see it has really helped us even when you reach home and there is nothing, the parent does not have money, you can buy yourself sandals and life goes on well*' (22E Px)

'*This programme really helps us when they give us money every term which makes us stop longing for bad behaviours in order to get that money. . ...you don't have to sleep with a man to get how to survive*' (21E P4)

'*Yes, there are still those who have boyfriends, you find the more you get the more you want to have more, and think of having big things, if you can buy this, you will say let me go to so and so to help me get a better one*' (21E P4)

Opinion was strong that receipt of the CT intervention had reduced school dropout, and was in stark contrast to previous years, as well as in other (non-participating) classes at the same schools. This, some felt, resulted from encouragement and motivation from being a study participant. Others opined that it stemmed from the ability to purchase items that helped girls remain in school. Others still voiced that there was a decrease in pregnancy due to reduced sexual activity resulting from less need to have transactional sex. The dialogue below captures views apparent across this group.

'*Since we started CCG school dropout has really reduced and even it is not there'(P1) (interrupting) 'Its even zero. Compared to last year. Since last year, many girls in form 4 for instance*

*were just pregnant. They got pregnant with the boda boda people or people's husbands so this year such cases are not there'* (P2)

*'Why is this*?' (Moderator)

'*Because we have our own money so we can depend on ourselves even now we cannot be lied to by those boda boda guys or our school mates to make us drop out of school'* (P1)

'*Because of the lessons, when we meet with you, you teach us how to take care of ourselves* (P2) '*to avoid relationships'*(Px) (All 22E)

## Girls in the combined CT and Cup group

As with the other study groups, some girls acknowledged that being educated in hygiene and puberty and to a lesser extent knowing HIV status gave confidence in talking to boys/men. It also assisted refusal of sexual advances.

'*Me, I can say that the behaviours have changed because the old days the rate of HIV and AIDS was very high but when you came we got teachings and got to know the disadvantages of HIV/AIDS so many girls try to distance themselves from the abuse and sexual behaviours* (23E Px)

There were striking similarities between the opinions and experiences of girls in the combined cash and cup group with the CT only group. This included a few girls admitting they did not like or use the cup. One group in particular seemed not to use it, using their CT to buy pads instead.

'*I tried to insert one day. I was harmed [giggling]. I removed immediately'* (16M Px)

Girls repeatedly acknowledged that relationships had altered primarily because the need for transactional sex was reduced. It was, however, more common for girls to ascribe this to the cash transfer rather than the cup, although both were mentioned as contributors.

'*Me, I can say this money has really helped us because earlier on we didn't even have things like panties, like biker [type of pant] or a bra so maybe you will say that I just want to look for a boy who can buy me all these things. But since we were given the money we stopped engaging with boys'* (23E P11)

'*Most schoolgirls what they normally want from boys is money, you will hear them making calls that 'have started my periods and I don't have pads', so now they cannot say they don't have pads because they have the mooncup'* (15M P3)

Although most girls believed transactional sex had reduced during the study, a few admitted that it continued. As one girl admitted '*They are not satisfied with the amount the study is giving. So they want to look for more'* (16M Px). Another girl spoke of hearing that other girls had to give money to the family so were not left enough for their own needs, hence required still more. A third participant admitted that multiple partners were still needed to satiate the requirement or desire for a range of necessities. Overall, many girls appeared to enjoy being free of these relationships, relishing independence and the choice of being able to say 'no'.

There was little mention about any changes in school attendance in the combined CT and cup FGDs. and no agreement as to whether the cup or the CT had the greatest impact on this.

'*I prefer cup because that cup we will use it after form 4 and it stays for long but this cash it can end*' (P2)

'*Me, I prefer cash because that menstrual cycle is only once per month, right*?' (Px)

'*I think it's the combination of both because the cup is used during menstruation but the money when you get it, it can be used for doing different things, you can even buy school uniform if you don't have, you can use it to buy socks if you lack it, or school shoes*' (P1) (All 24E)

## Discussion

This qualitative study explores girls' observations of the impact that participation in an intervention study has on their schooling and relationships. CT with or without a menstrual cup, was perceived to have the strongest effect on outcomes, namely some avoidance of pregnancy and consequent school dropout, either directly through payment of fees or the purchase of school-related goods, or indirectly. This was by having the means to purchase necessities, thus reducing the need for transactional sex. We noted that girls focused on the reduction of pregnancy as an outcome of abstaining from transactional sex, with seemingly little recognition given to risk reduction of HIV or STIs. Yet some were appreciative of knowing their HIV status. Receipt of a menstrual cup only was not recognised as having such a strong effect on pregnancy and dropout, although it was perceived as reducing *some* transactional sex. However, it was strongly attributed to impacting on school attendance. Emerging inductively from the narratives, the non-material inputs to the study, comprising aspects such as puberty and hygiene education, counselling, and motivation, were noted as beneficial, most strongly by girls in the control and cup groups. These encouraged girls to remain in school or bestowed some degree of confidence or self-efficacy facilitating refusal of unwanted sex. However, girls in our study did not recognise these skills as having as strong an effect on school retention as poverty, which was particularly seen as a greater driving force. However, they appeared to be greatly valued, boosting girls' sense of empowerment.

For the most part, girls perceived CT alone or with a cup as having a noticeable effect on school retention. They suggested this occurred through two means; monies received contributed directly to school levies or to the purchase of items facilitating school attendance and performance, e.g. pens, books, uniform, underwear. It also reduced the need for transactional sex to purchase these along with other necessities. Furthermore it enabled cash payment for travel to and from school with boda boda drivers. Decreasing sexual activity was perceived generally to reduce risk of pregnancy, which participants perceived led to school dropout. Some girls in the CT groups reported ceasing transactional sex altogether as the monies provided by the study met their needs (although many others stated it was not enough). Stoebenau et al [38] reports three key paradigms to transactional relationships that operate on a continuum; '*sex for basic needs*' '*sex for improved social status*' and '*sex and material expressions of love*'. A number of studies describe the giving of money or gifts for or following sex as providing a token of appreciation, reciprocity, or equity in the relationship [39–41] rather than a coercive act [42]. Girls or women are seen as having agency in the relationship [39, 40, 43] where sex is driven by '*wants*' rather than survival needs. In contrast, we suggest that poverty underpinned these relationships for our participants. '*Sex for basic needs*' i.e. a survival strategy, appeared the most important driver for our participants, in agreement with studies by Oruko et al [18] and Kyegombe et al [44]. Our supposition is further evidenced by the pleasure some girls in receipt of CT spoke of in now feeling able to refuse sex. This suggests a lack of choice previously. Saying 'no' to boys and men appeared to give these participants some new found degree of sexual

autonomy [45]. Oruko et al [18] describe schoolgirls in Siaya as feeling forced or obliged to have sex; our baseline survey reveals that of 1090 girls who had ever had sex, 82.3% had not wanted to, whilst 54.4% stated their first sex was forced [36]. The prior lack of sexual autonomy, and thus the importance for behaviour change, is further demonstrated by some recognition that men, (including boda boda drivers), did not approach them if they knew girls were in CT schools. This highlights men's role as predator, targeting girls whom they know lack financial resources and are particularly vulnerable to sexual acquiescence in return for money or gifts.

Despite receiving regular cash (with or without a menstrual cup), some girls reportedly continued engaging in transactional sex. It has been documented that adolescents succumb to peer pressure by acquisition of material possessions, or having a succession of boyfriends or sponsors to bestow a feeling of self-worth [46]. We speculate some participants may have been lured into continuing this practice to be rewarded by 'improved social status' as described above [38]. Alternatively, some participants may have needed more resources than were provided by the study, as some girls themselves stated. With chores, long distance travelling to school, and homework, girls reported little time to earn money even if opportunities were available. The baseline FGDs, along with our previous research in Siaya, highlight that some parents/guardians 'turn a blind eye' or actively encourage their daughters/wards to have 'boyfriends', either to save the householder from having to support them, or because they need to purchase food or fuel, or to put siblings through school etc [18, 19, 47].

Girls perceived that having both cash and a cup considerably reduced engagement in transactional sex. However, their narratives hardly appeared qualitatively different from those girls receiving CT only. Some girls told us that they chose not to use their cup, instead using some of their cash to purchase sanitary pads, which may explain similarity in narratives. From a programming perspective, the additional costs, delivery, and training of providing a cup in addition to CT seem to provide minimal add-on benefit. A CT only programme would therefore appear more sustainable.

Unlike girls in CT groups with or without the cup, girls who only received a menstrual cup felt that it had a minimal effect on the risk of pregnancy and subsequent dropout. However, the cup was frequently perceived to reduce school absenteeism. This was despite, as reported above by the combined group, some girls chosing not to wear it at all, or reported hesitancy in using it. Both factors have been reported elsewhere [48–50]. Many qualitative studies document that girls absent themselves from school during their menses for a variety of reasons including menstrual pain, having suitable WASH facilities or time to change their absorbent [20, 23, 24, 51]. Lacking a reliable and comfortable method of menstrual protection is, however, also a key factor linked to school absenteeism as girls fear leakage, odour and period teasing. This leads to difficulties in concentration [18, 19, 24, 52]. Providing a suitable alternative had the perceived effect of minimising these hurdles and facilitating greater attendance. Thus it presents further evidence in the qualitative versus quantitative debate over the role menstrual products play in reducing school absenteeism.

Our qualitative data has demonstrated that study participants perceived additional benefits from study participation over and above receipt of cash and / or a menstrual cup. Feeling valued and worthy through study inclusion, and receiving new and vital knowledge, particularly on SRH risks were noted gains. Being able to refuse sex appeared to be relished by girls receiving cash, suggesting it gave them confidence and a sense of empowerment [53]. This affords greater control of their own future. Improving attributes such as self-esteem has been linked with a lower risk of engaging in risk behaviours and experiencing social problems, and better mental health [54–56]. The importance of gaining self-esteem, sexual autonomy or empowerment cannot be overstated in this male dominated setting. Here dependence on boys and men

is the social norm [18];the boy child is favoured, fees for schooling are apportioned more to boys, and their sisters are sent home for non-payment [57]. Oruko et al. [18] reported a number of secondary school girls in the same setting, spoke about parental absence or neglect, and our baseline FGDs also suggested some general lack of concern for daughters' welfare, nurturing feelings of low self-worth. Studies that incorporate skills, health education or empowerment have been highly effective in helping to address gender norms, and protect girls from SRH harms [5, 58] The narratives emerging inductively from our discussions suggest girls from backgrounds such as rural Kenya would both welcome and benefit from such input.

## Study strengths and limitations

Girls openly talked about their relationships, including those of a transactional nature, suggesting that this is the norm within the wider community. The openness of discussion suggests that we were able to tap into the experiences of our participants. It may be, however, that this was also influenced by the gratitude girls felt in being part of a research study whereby they received, at minimum, puberty and hygiene education. This may also have contributed to response bias whereby the participants say what they believe the researchers want them to say. However, much of the narrative pertaining to empowerment and thoughts around the interventions and their effects emerged quite naturally through the discussions, as we concentrated initially on questioning around the material intervention and its influence on relationships and school outcomes. That some differences appeared to exist according to the different interventions provides a degree of confidence that social desirability responses may have existed but were not dominating discussions. As is the nature of FGDs, findings may have been skewed by dominance of certain members of each group, and other members not wanting to contradict others in the discussion. Further, because girls did not always clearly articulate their Px and we did not wish to halt the flow of dialogue by stopping girls and asking for more precision on their ID etc. the transcriber was unfortunately unable to assign a number of quotes to individuals (as denoted by Px in the paper). This limited our analysis in looking at effects of age and SES on perceptions and behaviours. We are confident that the limitations suggested were minimised by setting ground rules to begin each discussion. Analysis was undertaken in a reflexive manner, with constant checking to ensure that the information portrayed was not biased but appeared to represent the group narratives. As a qualitative study, we must also acknowledge that these findings were our interpretation of girls' perceptions and cannot therefore be presented as 'fact'.

## Conclusions

The provision of CT was perceived by recipients to reduce some transactional sex, thereby lowering pregnancy and subsequent school dropout. However, there was little qualitative evidence from girls in the menstrual cup group that its provision had a similar perceived effect, although seemingly reducing levels of absenteeism. This remains to be ascertained in the randomized controlled trial outcomes. Although girls receiving a menstrual cup spoke of the positive impact on school attendance, they felt many of their additional material needs were not met; accordingly transactional sex remained a necessary recourse, albeit less frequently. Consequently, school dropout appeared little altered to this group. The control group ultimately perceived little behaviour change. However, they (along with girls from the other groups) seemingly derived benefits from study inclusion including the puberty and hygiene education. We suggest future programmes consider alleviating material deprivation through schemes such as CT, alongside inclusive counselling and education packages. This has the potential to empower girls in refusing unwanted sex, and in understanding SRH risks, with the added

benefit of motivating academic achievement, ultimately leading to the completion of school education.

## Supporting information

**S1 Checklist. Inclusivity in global research.**
(DOCX)

**S1 Fig. Supplementary information flowchart of randomisation and qualitative study design.**
(DOCX)

**S1 Table. Supplementary information 1.1 baseline FGD.**
(DOC)

**S2 Table. Supplementary information 1.2 midline FGD.**
(DOCX)

**S3 Table. Supplementary information 1.3 endline FGD.**
(DOCX)

## Acknowledgments

We would like to thank the girls, schools, community, and the Ministry of Health and Education partners for their support and participation in this study. We are grateful to the field staff, including the moderators and note takers who conducted the discussions, and staff from Safe Water and AIDS Programme who distributed the intervention, for their contribution to this study. The KEMRI Director approved publication of this paper.

## Author Contributions

**Conceptualization:** Linda Mason, Penelope Phillips-Howard.

**Data curation:** Linda Mason, Garazi Zulaika, Elizabeth Nyothach.

**Formal analysis:** Linda Mason.

**Funding acquisition:** Linda Mason, Anna Maria van Eijk, Penelope Phillips-Howard.

**Investigation:** Linda Mason, Garazi Zulaika, David Obor, Elizabeth Nyothach.

**Methodology:** Linda Mason, Garazi Zulaika.

**Project administration:** Linda Mason, Garazi Zulaika, Penelope Phillips-Howard, Elizabeth Nyothach.

**Resources:** Eunice Fwaya, David Obor, Elizabeth Nyothach.

**Software:** Elizabeth Nyothach.

**Supervision:** Linda Mason, Penelope Phillips-Howard.

**Validation:** Garazi Zulaika, Anna Maria van Eijk.

**Writing – original draft:** Linda Mason.

**Writing – review & editing:** Garazi Zulaika, Anna Maria van Eijk, Eunice Fwaya, Penelope Phillips-Howard, Elizabeth Nyothach.

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
