## [Decision Letter · Decision Letter 0]

6 Jun 2022

PGPH-D-22-00265

‘You don’t have to sleep with a man to get how to survive’: Girl’s perceptions of an intervention study aimed at improving sexual and reproductive health and schooling outcomes.

Dear Dr. Mason,

Thank you for submitting your manuscript to PLOS Global Public Health. After careful consideration, we feel that it has merit but does not fully meet PLOS Global Public Health’s publication criteria as it currently stands. Therefore, we invite you to submit a revised version of the manuscript that addresses the points raised during the review process.

Based on the reviewers' comments, you will need to clarify the manner through which the interventions were administered and elaborate more extensively on the sampling approach and focus group discussion procedure used.

Please submit your revised manuscript by . If you will need more time than this to complete your revisions, please reply to this message or contact the journal office at globalpubhealth@plos.org. Please include the following items when submitting your revised manuscript:

We look forward to receiving your revised manuscript.

Kind regards,

Adriana Andrea Ewurabena Biney

Academic Editor

Journal Requirements:

1. Please include a complete copy of PLOS’ questionnaire on inclusivity in global research in your revised manuscript. Our policy for research in this area aims to improve transparency in the reporting of research performed outside of researchers’ own country or community. The policy applies to researchers who have travelled to a different country to conduct research, research with Indigenous populations or their lands, and research on cultural artefacts. The questionnaire can also be requested at the journal’s discretion for any other submissions, even if these conditions are not met.  Please find more information on the policy and a link to download a blank copy of the questionnaire here: https://journals.plos.org/plosone/s/best-practices-in-research-reporting. Please upload a completed version of your questionnaire as Supporting Information when you resubmit your manuscript.”

- State the initials, alongside each funding source, of each author to receive each grant.

Additional Editor Comments (if provided):

Based on the reviewers' comments, you will need to clarify the manner through which the interventions were administered and the FGD method used.

Regarding the sample selection, additional information on selection of girls from the sampled schools is required. Understanding the inclusion criteria, what approach was used to invite the girls to participate, and who and how specifically were they targeted?

A rationale for the use of FGDs as opposed to other qualitative methods is also required.

You will also need to respond to reviewers' critical concerns about respondent bias and correct minor grammatical errors.

Reviewers' comments:

Reviewer's Responses to Questions

**Comments to the Author**

1. Does this manuscript meet PLOS Global Public Health’s publication criteria? Is the manuscript technically sound, and do the data support the conclusions? The manuscript must describe methodologically and ethically rigorous research with conclusions that are appropriately drawn based on the data presented.

Reviewer #1: Yes

Reviewer #2: Partly

2. Has the statistical analysis been performed appropriately and rigorously?

Reviewer #1: N/A

Reviewer #2: N/A

3. Have the authors made all data underlying the findings in their manuscript fully available (please refer to the Data Availability Statement at the start of the manuscript PDF file)?

Reviewer #1: No

Reviewer #2: Yes

4. Is the manuscript presented in an intelligible fashion and written in standard English?

Reviewer #1: Yes

Reviewer #2: Yes

5. Review Comments to the Author

Reviewer #1: I have attached the comments to author. Comments to the Authors

Thank you for the opportunity to read the following manuscript: “‘You don’t have to sleep with a man to get how to survive’: Girl’s perceptions of an intervention study aimed at improving sexual and reproductive health and schooling outcomes.” This is an important study on the perspectives of girls and women who participated in a study on the effectiveness of cash transfers and menstrual cups to improve sexual and reproductive health. Overall, I think this manuscript shares important information not often collected from trials, but the paper can be improved.

Abstract:

In the background section, could you include some information on factors known to be associated with your three main outcome variable (pregnancy, drop out, absenteeism)? Please include more information on the study population (e.g., age range, what country, what SES, other important descriptive variables) in the methods section. Tell us more about the FG methods (were the interviews semi-structured – when was baseline, midline, and study end completed)? What was the study duration?

General Language/Formatting:

I believe it is sub-Saharan (not Sub-Saharan), Also, your first LMIC abbreviation should include an ‘s’ as should STI’s and RTI’s. Also, “decision making at THE household and employment level”. Enrollment in line 98 is spelled wrong. Enroll in line 98 is spelled wrong. The comment that 4/10 girls complete secondary education needs a citation. Enrollment is spelled wrong throughout the manuscript and there are several typos. This sentence needs a citation: “Girls in rural Kenya report engaging in transactional sex to acquire necessities, including to fund their schooling needs.” Data is plural, so please change ‘data was’ to ‘data were’ and so forth throughout. I recommend using the terms boys and girls or men and women instead of sex-specific terms like male and female. The three supplementary files are not referenced in the text, but should be.

Introduction:

AGYW from SSA endure 2x the rate of morbidity and mortality compared to any other region? Do you mean here that AGYW in SSA have 2x the rate of M&M compared to AGYW outside SSA? Is this the overall M&M rate or related to SRH health risks? For ‘better employment chances” do you mean opportunities for employment? There is complexity in poverty, retention in school, and pregnancy. Does poverty drive early marriage? Do schools allow girls to remain in school if pregnant? Some of these issues could be better disentangled or at least described in the introduction.

Methods:

What does ‘issued termly’ mean? How was ‘80% school attendance’ measured? What was the general age range, as secondary school age can differ by educational system? Were the same girls retained in each focus group for all three time points?

Results:

Please provide more interpretation or context before and after quotes. Why did the person say she couldn’t go back to school in the 3rd quote? Weren’t all girls in this study in school (due to the selection of randomizing schools)? Why were the mid and endline results presented in the same part of the results section? Was there no difference between immediately after and 2 years after the intervention?

Discussion:

Selection bias likely existed since all participants were selected from schools, therefore they attended school. You are likely missing the most vulnerable because of this selection process. This study is also vulnerable to response bias, meaning the participants may have responded to what they thought the researchers wanted to hear.

Reviewer #2: This manuscript presents an innovative use of experimental design and qualitative techniques in exploring sexual and reproductive health issues among adolescent girls. The manuscript is well written and the issues/findings are sufficiently highlighted. The finding that cash transfers seem to make the most impact compared to the provision of menstrual cups only or in addition to cash is an interesting finding. That said, some aspects of the manuscript, particularly the methods section needs some further details. See further comments below.

1. It appears the pre-intervention interviews explored the questions of interest among girls in general before any intervention/treatments were applied. If so this needs to be clearly spelt out.

2. How exactly were the interventions/treatments administered? Were the interventions e.g. cash transfer administered to selected girls in particular schools or to all girls in selected schools? How were the girls who were selected for the FGDs subsequently selected? For example did girls or schools who received menstrual cups only constitute one homogenous group (Cup only) for the FGDs? it is important that these details are provided to guide the reader.

3. The information provided in Table 1 seems to show that different schools and different girls were interviewed in the pre-intervention, mid-line and end-line focus group discussions. The table presents 24 different focus groups for the collective total of 24 FGDs that were conducted for this study. It is, however, not clear if the same group of girls were interviewed in the same groups at baseline, midline and end line or different groups of girls were interviewed at the different time points.

It will be helpful if the authors can provide a diagram or figure that shows; (1) how the girls were randomly assigned to the 4-arms, and (2) how the various treatments were assigned, and (3) how the FGDs were conducted according to treatment arms (for the 24 FGDs reported in Table 1).

4. What informed the choice of menstrual cups rather than other menstrual products such as pads or tampons. While menstrual cups tend to be long lasting and reusable, usage is more difficult and less common. This could inform the findings of low use and non-use reported by the girls. Also, were the girls educated on how to use the menstrual cups as part of the study?

Given that menstrual cups seem to have made minimal impacts, would the authors recommend the provision of pads for example as this seems to be more common and user friendly for the girls?

5. The authors mention using thematic analysis but the presentation of the results does not seem to follow particular themes. Rather the results are presented more along the lines of issues discussed in the various FGDs.

Minor formatting issues

1. Page 25, line 606 - Delete while

2. Page 27, line 647 - write IDetc as ID etc.

6. PLOS authors have the option to publish the peer review history of their article (what does this mean?). If published, this will include your full peer review and any attached files.

**Do you want your identity to be public for this peer review?** For information about this choice, including consent withdrawal, please see our Privacy Policy.

Reviewer #1: No

Reviewer #2: No

---

## [Decision Letter · Decision Letter 1]

13 Sep 2022

‘You don’t have to sleep with a man to get how to survive’: Girl’s perceptions of an intervention study aimed at improving sexual and reproductive health and schooling outcomes.

PGPH-D-22-00265R1

Dear Dr Mason,

We are pleased to inform you that your manuscript '‘You don’t have to sleep with a man to get how to survive’: Girl’s perceptions of an intervention study aimed at improving sexual and reproductive health and schooling outcomes.' has been provisionally accepted for publication in PLOS Global Public Health.

Before your manuscript can be formally accepted you will need to complete some formatting changes, which you will receive in a follow up email. A member of our team will be in touch with a set of requests. In addition to the formatting requests, please  perform the grammatical corrections noted by the reviewer below.

Best regards,

Tia M. Palermo

Academic Editor

Reviewer Comments (if any, and for reference):

Reviewer's Responses to Questions

**Comments to the Author**

1. If the authors have adequately addressed your comments raised in a previous round of review and you feel that this manuscript is now acceptable for publication, you may indicate that here to bypass the “Comments to the Author” section, enter your conflict of interest statement in the “Confidential to Editor” section, and submit your "Accept" recommendation.

Reviewer #2: All comments have been addressed

2. Does this manuscript meet PLOS Global Public Health’s publication criteria? Is the manuscript technically sound, and do the data support the conclusions? The manuscript must describe methodologically and ethically rigorous research with conclusions that are appropriately drawn based on the data presented.

Reviewer #2: Yes

3. Has the statistical analysis been performed appropriately and rigorously?

Reviewer #2: N/A

4. Have the authors made all data underlying the findings in their manuscript fully available (please refer to the Data Availability Statement at the start of the manuscript PDF file)?

Reviewer #2: No

5. Is the manuscript presented in an intelligible fashion and written in standard English?

Reviewer #2: Yes

6. Review Comments to the Author

Reviewer #2: The authors have adequately addressed comments from an earlier review of the manuscript. There are however a few grammatical errors and omissions that the authors need to pay attention to.

For example.

Line 39 include "that" between found and narratives

Line 104: include period after reference number 11

Line 105: include period after reference number 12

Line 211: insert to between agree and a

Line 248: Write 231 in words

Line 548: delete girls between repeatedly and acknowledged

7. PLOS authors have the option to publish the peer review history of their article (what does this mean?). If published, this will include your full peer review and any attached files.

**Do you want your identity to be public for this peer review?** For information about this choice, including consent withdrawal, please see our Privacy Policy.

Reviewer #2: No
